# Multilevel Heterogeneity of Colorectal Cancer Liver Metastasis

**DOI:** 10.3390/cancers16010059

**Published:** 2023-12-21

**Authors:** Hao Chen, Chongya Zhai, Xian Xu, Haidong Wang, Weidong Han, Jiaying Shen

**Affiliations:** Department of Medical Oncology, Sir Run Run Shaw Hospital, School of Medicine, Zhejiang University, Hangzhou 310016, China; 22218189@zju.edu.cn (H.C.); yu1988410@sina.com (C.Z.); 11918538@zju.edu.cn (X.X.); haidongwang@zju.edu.cn (H.W.)

**Keywords:** colorectal cancer liver metastasis, heterogeneity, gene, transcriptome, protein, metabolism, immune, therapy

## Abstract

**Simple Summary:**

Liver metastasis remains a major hurdle to the long-lasting survival of patients with colorectal cancer (CRC). When CRC spreads to the liver, tumor cells undergo a series of changes to adapt to the new environment, leading to evident heterogeneity. Therefore, understanding the features of metastatic tumor cells in the liver is valuable for the overall control of patients with colorectal cancer liver metastasis (CRLM). In this review, we provide a comprehensive overview of the spatial heterogeneity of CRLM at different molecular levels, including genetics, transcriptomics, proteomics, metabolism, and immune levels. While genetic heterogeneity may not be apparent, the other four levels demonstrate significant heterogeneity. Compared to primary CRC, the epithelial–mesenchymal transition (EMT)-related proteins are significantly altered. Moreover, the metabolic activity is enhanced and the infiltration of immuno-suppressive cells is increased in CRLM. All these changes promote the metastasis and colonization of CRC cells. Based on these findings, we also summarize preclinical therapeutic modalities targeting the heterogeneity of CRLM, aiming to provide new directions for clinical interventions and improve the survival rates of these patients.

**Abstract:**

Colorectal cancer liver metastasis (CRLM) is a highly heterogeneous disease. Therapies that target both primary foci and liver metastasis are severely lacking. Therefore, understanding the features of metastatic tumor cells in the liver is valuable for the overall control of CRLM patients. In this review, we summarize the heterogeneity exhibited in CRLM from five aspects (gene, transcriptome, protein, metabolism, and immunity). In addition to genetic heterogeneity, the other four aspects exhibit significant heterogeneity. Compared to primary CRC, the dysregulation of epithelial–mesenchymal transition (EMT)-related proteins, the enhanced metabolic activity, and the increased infiltration of immunosuppressive cells are detected in CRLM. Preclinical evidence shows that targeting the EMT process or enhancing cellular metabolism may represent a novel approach to increasing the therapeutic efficacy of CRLM.

## 1. Introduction

Colorectal cancer (CRC) is a disease with a high incidence and mortality rate. The number of new CRC cases worldwide reached 1.93 million in 2020, ranking third after breast cancer and lung cancer. The number of CRC-related deaths reached 940,000, making it the second most deadly tumor globally [1]. In China, according to the 2016 national cancer statistics published by the National Cancer Center, a total of 4.06 million tumor patients were diagnosed in 2016, with approximately 408,000 being CRC patients. Among these cases, approximately 196,000 CRC patients died, accounting for 8.10% of the total [2].

The high metastasis rate of CRC is one of the reasons for its high mortality rate [3,4]. CRC frequently metastasizes, particularly to the liver, with approximately 20% of patients having liver metastasis by the time CRC is diagnosed. This significant rate of colorectal liver metastasis (CRLM) reduces the effectiveness of treatment for CRC patients [5,6]. These patients are often diagnosed with advanced tumors, and their disease is poorly controlled. In addition, the available treatment options for CRLM patients are relatively limited. Surgical resection is the primary treatment for CRLM patients, but only a small proportion can be cured by resection of liver metastasis, and the prognosis of other patients is poor [7,8]. Immunotherapy has shown notable effectiveness in mismatch-repair-deficient and microsatellite instability-high (dMMR–MSI-H) CRC patients [9,10,11]. However, the efficacy of immunotherapy in CRLM patients is limited, possibly due to the immunosuppressive microenvironment of the liver [12].

Tumor heterogeneity has a significant impact on the treatment outcome of tumor patients [13]. Although tumor heterogeneity is complex, the study of tumor heterogeneity remains a hot topic. Tumor heterogeneity can be classified into intratumor heterogeneity and inter-tumor heterogeneity, as well as temporal heterogeneity and spatial heterogeneity. Temporal heterogeneity refers to changes in the nature of tumors over time, while spatial heterogeneity indicates differences in the nature of different cell subpopulations within a tumor at the same site, as well as differences between the primary tumor lesion and its corresponding metastasis.

CRC is a highly heterogeneous disease, especially after CRLM. The unique microenvironment of the liver contributes to CRC exhibiting spatial heterogeneity in various aspects, including gene expression, tumor microenvironment, and biological behavior [14]. The high heterogeneity in CRLM is one of the major reasons for its poor response to antitumor treatment [15]. To improve the overall control of CRLM patients, it is important to have a comprehensive insight into the heterogeneity between primary CRC and CRLM. In this article, we summarize the heterogeneity between primary CRC and CRLM at the genetic, transcriptional, protein, metabolic, and immune levels and discuss the prognostic value of this heterogeneity and its impact on clinical decision making.

## 2. Genetic Heterogeneity

The adenoma–carcinoma sequence underlies the development of CRC and involves many changes, including tumor suppressor gene inactivation (APC, TP53), oncogene activation (BRAF, PI3KA, and RAS), chromosomal instability (CIN), CpG island methylation phenotype (CIMP), and microsatellite instability (MSI) pathways [16,17]. These alterations result in a highly unstable genome in CRC. However, when exploring the sources of genetic heterogeneity in CRLM, we were surprised to find little genetic heterogeneity between primary CRC and CRLM.

### 2.1. Key Driver Genes

There are five key driver genes, APC, TP53, RAS, BRAF, and PIK3CA, that play a critical role in the adenoma–carcinoma sequence of CRC, and the mutational status of these genes can influence the clinical outcome of CRC patients. Patients with KRAS wild-type tumors had significantly better clinical outcomes than patients with KRAS mutation tumors [18]. Patients with BRAF mutation and PIK3CA mutation tumors also showed poorer clinical outcomes [19,20]. Therefore, we summarize the heterogeneity of these five genes among CRLM patients. These genes did not show heterogeneity between the primary tumor and the corresponding liver metastasis [21,22,23,24,25]. Brunsell et al. examined mutation sites in KRAS, BRAF, and PIK3CA in 20 CRLM patients and found the same mutation status in 18 patients with primary tumor and liver metastasis [21]. Hou et al. investigated the heterogeneity of the KRAS pathway in CRC and found that the frequency of KRAS mutations was significantly higher in the lung (62.0%) and brain (56.5%) than in the liver (32.5%). Thus, KRAS mutation could be an independent predictor of lung metastasis but played a less significant role in CRLM [26]. Another study reached a similar conclusion [27]. In addition to the five genes mentioned above, SMAD4 is a key regulator of the TGFβ signaling pathway, which inhibits cell proliferation, promotes apoptosis, and regulates epithelial–mesenchymal transition (EMT) [28]. Itatani et al. found that the knockdown of SMAD4 led to a significant increase in the expression of CCL15, which led to the aggregation of CCR1+ myeloid cells in hepatic metastases and contributed to CRC cells colonizing the liver in the early stages of liver metastasis [29]. Although deletion of the SMAD4 gene is associated with tumor progression, metastasis, and poor prognosis, two other studies concluded that the mutation profiles of SMAD4 are highly concordant in primary CRC and CRLM [24,30].

### 2.2. Chromosomal Instability (CIN)

CIN is a common feature of solid tumors, including CRC, and causes genomic instability in approximately 70% of CRC patients [16,31]. CIN includes instability of the chromosome number (numerical CIN) and instability of the chromosome structure (structural CIN); numerical CIN refers to the increase or decrease of chromosome copy number; and structural CIN includes deletions, translocations, and derivative chromosomes, among others [32].

Previous studies found that chromosome instability is significantly higher in metastatic breast cancer cells than in primary cells, and it is also a driver of metastasis [33]. In CRC, some studies reached the same conclusion [34,35,36,37]. Mamlouk et al. found through whole-genome sequencing that the CNVs of the MMP9 and CDX2 genes were significantly increased in CRLM [37]. MMP9 belongs to the matrix metalloproteinase family, which can degrade various protein components in the extracellular matrix (ECM) and disrupt the histological barrier that prevents tumor cell invasion, and therefore plays a key role in tumor invasion and metastasis [38]. CDX2 is involved in the proliferation and differentiation of intestinal epithelial cells [39].

Nonetheless, some studies have come to the opposite conclusion. Leonie used a high-resolution array of comparative genomic hybridization to study 62 primary colorectal cancers and 68 matched metastatic lesions (22 liver, 11 lung, 12 ovary, 12 omentum, and 11 distant lymph nodes). They found that patterns of DNA copy number aberrations were highly similar between all primary and metastatic lesions [40]. Through allelic copy number analysis of 33 CRC samples, Shogo identified several chromosomal aberrations common in CRC patients, with gains on 20p13-p12.1 and 20q11.21-q13.33 and loss of heterozygosity (LOH) on 6q14.1-q25.1 more common in CRLM patients. Through genetic analysis of metastatic lesions, they found that allelic imbalances in CRLM were very similar to those in primary CRC and these aberrations on chromosomes 20p, 20q, and 6q were also present in CRLM, which suggests that they may promote CRLM [41]. Previous studies have also shown that only a few mutations are needed to transform highly aggressive tumor cells into metastatic ones [42]. These results indicate that CRC cells maintain relative chromosome stability during metastasis.

### 2.3. Microsatellite Instability (MSI) Status

Approximately 15% of CRC patients are affected by MSI pathways [10]. MSI is caused by functional defects in genes such as DNA mismatch repair genes (hMSH2, hMLH1, hMSH3, hMSH6, hPMSH1, and hPMSH2). Currently, there are two main methods to detect MSI status as follows: (1) immunohistochemistry (IHC), which detects the expression of four mismatch repair proteins (MLH1, PMS2, MSH2, and MSH6) in the nucleus to detect the presence of mismatch function defects, and (2) molecular testing, which detects the length of microsatellite sequences in tumor tissue to determine whether MSI is present at the site. Through IHC and molecular assays, current studies found a very high similarity of MSI status between primary CRC and CRLM [43,44,45,46]. Among them, He et al. and Jung et al. found partial differences, but the differences were concentrated in peritoneal and ovarian metastasis, and no differences were found in CRLM [43,45].

## 3. Transcriptomic Heterogeneity

### 3.1. MicroRNAs (miRNAs)

MicroRNAs (miRNAs), which are the most studied class of noncoding RNAs, are a class of short RNA molecules that range from 19 to 25 nucleotides and are primarily responsible for regulating posttranscriptional gene expression [47,48]. MiRNAs have been linked to many diseases, including CRC. In addition, they are involved in colorectal carcinogenesis and can be used as a marker for CRC metastasis [49,50].

After CRC metastasizes to the liver, different types of miRNAs enable cancer cells to adapt to the new environment of the liver by regulating the expression of their respective target genes. Using genome-wide expression profiling, Vychytilova-Faltejskova et al. identified that miR-143, miR-10b, and miR-28-5p were downregulated, while miR-122, miR-122*, and miR-885-5p were upregulated in liver metastases compared to their primary tumors [51]. Hur et al. and Zhang et al. reached the same conclusion [52,53]. MiR-122 is a liver-specific miRNA and a recognized suppressor of liver cancer. It exerts its effects by regulating the expression of important miRNAs in the liver and it has been shown to have a strong relationship with the prognosis of patients with liver cancer [54,55]. Another upregulated miRNA, miR-885-5P, promotes the proliferation and migration of CRC cells by stimulating the EMT pathway. Epithelial–mesenchymal transition (EMT) is a crucial first step in the process of tumor metastasis, and after tumor cells lose epithelioid features through EMT, the shed tumor cells metastasize to distant organs. Therefore, CRC patients with high miR-885-5p expression tend to have a worse prognosis [56,57]. However, research on miR-10b is still controversial. For example, several studies have confirmed that miR-10b is an oncogenic miRNA because its high expression is associated with worse outcomes [51,52,58]. Interestingly, Song et al. found that miR-10b inhibits the growth of CRC by regulating EMT in animal experiments [59]. There may be two reasons for the opposing conclusions. First, the organ characteristics of mice may be quite different from those of humans, and second, miRNAs may be responsible for regulating multiple mRNAs, which may have opposite effects. Therefore, the true role of miR-10b in CRC has not yet been determined.

In addition, Hur et al. found that the expression of miR-203 and miR-200c in CRLM was much higher than that in primary CRC [60,61]. MiR-200c promotes EMT mainly by suppressing the overexpression of target genes (ZEB1, ETS1, and FI1, three EMT-related genes) and therefore promotes the growth and metastasis of CRC. Hur also found that miR-203 is a secreted miRNA and metastatic lesions of CRC secrete miR-203 into circulation, which results in high serum miR-203; thus, high serum miR-203 is usually associated with distant metastasis. However, miR-203 is a potent tumor suppressor miRNA in many other tumors [62,63,64]. Torres et al. found significant upregulation of miR-424-3p, miR-503, and miR-1292 expression in CRLM, and these miRNAs might promote CRC metastasis [65].

### 3.2. circRNAs

Circular RNAs (circRNAs) are single stranded, covalently closed RNA molecules [66]. Initially, circRNAs were considered to be “junk” with little function [67]. However, with the development of certain technologies, such as immunohistochemistry (IHC) and high-throughput RNA sequencing (RNA-seq), it has been shown that circRNAs are involved in the development of many diseases [68,69]. Among them, circRNAs are mainly involved in the transcriptional regulation of genes by the following: (1) influencing the splicing of their linear homologs, (2) regulating the transcription of parental genes, and (3) acting as sponges for microRNAs and inhibiting their activity and, thus affect the expression of genes downstream of microRNAs [70,71,72,73,74,75,76]. Some circRNAs have also played a major role in CRC; for example, circ001971 and circ3823 can both promote tumor metastasis and angiogenesis [77,78,79]. Similar to miRNAs, the expression of some circRNAs changes in CRLM to promote tumor progression. Xu et al. analyzed three cases by RNA sequencing and found that 92 circRNAs were upregulated in CRLM compared to primary CRC and 21 circRNAs were downregulated in CRLM [80]. Among them, circRNA_0001178 and circRNA_0000826 were most significantly upregulated in CRLM and were considered promising markers of CRLM. In addition, Chen et al. and Zhang et al. found that circNSUN2 and hsa_circ_0006401 were also upregulated in CRLM, and promoted tumor progression [81,82]. CircNSUN2 is a m6A-modified circRNA and it forms a CircNSUN2/IGF2BP2/HMGA2 complex with insulin-like growth Factor 2 mRNA-binding protein 2 (IGF2BP2) and high mobility group AT-hook 2 (HMGA2). This complex could improve the stability of HMGA2 RNA and thereby increase the expression of the HMGA2 protein. As reported by Li et al., HMGA2 induces EMT and promotes CRC progression [83]. Chen RX also analyzed the changes in EMT-related proteins after circNSUN2 overexpression and found that the expression of the epithelial marker E-cadherin decreased and the expression of the mesenchymal marker vimentin increased. This further suggests that circNSUN2 can promote EMT in CRC cells through the HMGA2 pathway.

### 3.3. LncRNAs

Long noncoding RNAs (lncRNAs) are the third class of noncoding RNAs in addition to miRNAs and circRNAs. LncRNAs are a class of noncoding RNAs that are greater than 200 nt in length. LncRNAs play an important role in all levels of gene function and regulation. LncRNAs primarily interact with mRNA, DNA, proteins, and miRNAs to regulate gene expression at the epigenetic, transcriptional, posttranscriptional, translational, and posttranslational levels in a variety of ways [84]. For example, a recent study reported that when antisense lncRNAs were expressed, demethylation of protocadherin (Pcdh) DNA at this site led to the remodeling of the chromosome structure, which enabled the distal enhancer to bind to the promoter and promote the expression of the Pcdhα gene [85]. Great progress has been made in studying the role of lncRNAs in CRC. Nevertheless, only a few studies have examined the heterogeneity of lncRNAs in CRLM, and two lncRNAs associated with glucose metabolism (lncRNA GAL and lncRNA MIR17HG) have been found to be upregulated in CRLM [86,87]. Interestingly, the MIR17HG/miR-138-5p/hexokinase (HK1/2) pathway enhances glycolysis, and increased lactate (a metabolite of glycolysis) that activates the p38/ELK1 pathway; this promotes the expression of MIR17HG and, forms a positive feedback loop for promoting tumor invasion and metastasis.

### 3.4. Epigenetic Heterogeneity

Epigenetic modifications are genetic changes that occur by altering genomic DNA or chromatin structure and noncoding RNA modifications without altering the DNA sequence itself, and this mainly includes the following: (1) DNA methylation, which is often found on cpG islands—CpG Island Methylator Phenotype (CIMP), and (2) histone modifications, including methylation, acetylation, and phosphorylation. A large body of literature confirms that epigenetic modifications play a key role in the distant metastasis of CRC [88,89,90]. Chen et al. found that low expression of METTL4 (m6A writer) in CRC reduces m6A modification of the SRY-related high-mobility-group Box 4 (SOX4) gene, which prevents YTHDF2 (m6A reader) from being recognized and degraded by YTHDF2, while increased expression of SOX4 can promote CRC metastasis [91]. Therefore, greater understanding is needed regarding the specific epigenetic alterations during CRLM and their effects, which may guide the development of targeted and personalized therapeutic strategies.

Studies have found that epigenetic reprogramming occurs during CRLM. Teng et al. found that there are some unique super-enhancers in the vicinity of liver metastatic cells and two specific transcription factors in the liver (FOXA2 and HNF1A) can bind to these unique enhancers and reshape the epigenetic landscape in liver metastasis, which promotes the expression of certain liver-specific genes [92].

For DNA methylation, several papers have reported heterogeneity among CRLM. Miranda et al. found increased methylation of Ras association (RalGDS/AF-6) domain family member 1 protein (RASSF1a) in hepatic metastases, while cyclin-dependent kinase inhibitor 2A (p16INK4a) methylation decreased. These authors hypothesized that the hypermethylation of RASSF1a originated from contamination of adjacent nontumorigenic hepatocytes; whereas, p16INK4a is an important oncogene that inhibits cell-cycle progression, and when it is dysfunctional, it leads to uncontrolled proliferation of tumor cells [93,94]. M6A(N6-methyladenosine) modification is also a type of DNA methylation. We mentioned above the m6A-modified circNSUN2, which is highly expressed in CRLM and can exit the nucleus with the help of YTHDC1 (m6A reader) to promote CRC metastasis [81]. In addition, a m6A writer, METTL3, was found to be upregulated in CRLM, and the upregulation of METTL3 expression was associated with poor prognosis of patients because METTL3 can increase SRY (sex-determining region Y)-Box 2 (SOX2) methylation. This methylation can be recognized by IGF2BP2 (m6A reader), which promotes the expression of SOX2 downstream target genes (CCND1, MYC, etc.) and also promotes stemness and metastasis of CRC cells [95].

Protein acetylation is an epigenetic modification that plays an important role in the metastasis of CRC cells. However, there are relatively few studies on the heterogeneity of acetylated proteins. To fill this research gap, Shen et al. performed a comprehensive analysis of protein lysine acetylation in primary CRC and CRLM using mass spectrometry [96]. They found significant differences in the expression of acetylated proteins in primary CRC and CRLM. In addition, they evaluated the differences in lysine acetylation sites in these proteins. In contrast to primary CRC, 31 acetylation sites for 22 proteins were downregulated in CRLM, and 40 acetylation sites for 32 proteins were upregulated. Of these, acetylated histone H3.2 at Lys 19 (HIST2H3AK19Ac) and TPM2 K152Ac were the most significantly downregulated in CRLM, whereas acetylated histone H2B type 1-L at Lys 121 (H2BLK121Ac) and ADH1B K331Ac were upregulated. The distribution and function of these proteins were also summarized and most of the proteins were mainly distributed in cells, organelles, macromolecular complexes, and membranes. Furthermore, they were mainly involved in the carbon metabolism and biosynthesis of amino acids. Another study by Shen et al. was conducted on another protein, isocitrate dehydrogenase 1 (IDH1). This protein is highly acetylated in CRLM, and sirtuin-2 (SIRT2) can mediate the deacetylation of the IDH1 protein, which inhibits the proliferation and distant metastasis of CRC cells in vivo/ex vivo [97].

## 4. Protein Heterogeneity

### 4.1. EMT-Related Proteins

As previously discussed, EMT is an essential process in CRLM progression. In this process and due to the action of the involved proteins through signaling pathways, epithelial cells lose connection and apical-basal polarity, which enables tumor cells to acquire greater motility and enable metastasis. Among them were adhesion-related proteins (E-cadherin, N-cadherin, tight junction family proteins, etc.), and α-SMA, Snail, and Twist proteins play a major role in this process [98].

For migration-associated proteins, Yin et al. found that cell migration-related protein vitronectin (VTN) and actin-related protein (ARP3) expression was higher in CRLM than in primary CRC by large-scale quantitative proteomic analysis [99]. Using a similar approach, Liu et al. identified 311 proteins that were dysregulated in CRLM, including fibronectin 1 (FN1), tissue inhibitor of metalloproteinases 1 (TIMP1), Versican (VCAN), periostin (POSTN) and thrombospondin-1 (THBS1), which have been identified as the five most critical proteins that promote CRC metastasis [100]. For example, THBS1 promotes CRC metastasis by enhancing EMT; FN1, TIMP1, VCAN, and POSTN have all been shown to play a role in the process of CRC metastasis [101,102,103,104,105]. In addition, insulin-like growth factor binding protein 7 (IGFBP7) has been found to be downregulated in CRLM, which inhibits EMT to block CRC metastasis [106].

Decreased adhesion of cancer cells to each other and separation and detachment from the primary lesion is a key step in metastasis. Integrins are the major cell adhesion receptors and claudins are tight junction proteins [107,108] and both maintain adhesion between cells and prevent cancer cells from shedding. Yin et al. found that integrin alpha5 (ITA5) expression is decreased in CRLM [99]. Wang et al. and Georges et al. also found that claudin-1, claudin-4, and claudin-7 were downregulated in CRLM [109,110]. Another protein, Rho GTPase-activating protein 5 (ARHGAP5), was significantly upregulated in CRLM. ARHGAP5 is a GAP that regulates the Rho family of small GTPases, and researchers determined this by knocking down this protein, downregulating E-cadherin expression, upregulating N-cadherin and vimentin expression, and inhibiting CRC metastasis. It was further demonstrated that it could affect the invasion and metastasis of CRC cells by regulating the activity of EMT [111]. The decreased expression of these proteins reduces the adhesion between cells so that cancer cells can take the first step of metastasis.

The Wnt/β-catenin and MAPK signaling pathways are the most important pathways in CRC progression, and both of these pathways can promote EMT in CRC cells and thus promote CRC cell invasion and metastasis [112,113]. Tang et al. found that a protein involved in the MAPK signaling pathway, PEA15, was expressed significantly higher in CRLM than in primary CRC [114]. Phosphoprotein enriched in astrocytes-15 kDa (PEA15) can promote EMT by activating the MAPK signaling pathway. Two other proteins (ATP6L and FILIP1L) involved in the Wnt signaling pathway have also been found to be heterogeneous. ATP6L, the C subunit of the V-ATPase V0 domain, has previously been shown to enhance the invasion and metastasis of breast cancer cells in vitro [115]. Wang et al. demonstrated the role of ATP6L in CRC through in vivo experiments in mice [116]. ATP6L is needed for the activation of the Wnt/β-catenin signaling pathway, and it is also responsible for regulating the acidic tumor microenvironment, which could induce cancer cells to secrete proangiogenesis factors, such as interleukin-8 and vascular endothelial growth factor and is therefore beneficial to tumor angiogenesis and growth. Another protein, filamin A-interacting protein 1-like (FILIP1L), differs from ATP6L in that overexpression of FILIP1L in CRC cells inhibits the WNT signaling pathway and thereby inhibits EMT. Ku et al. used mass spectrometry to compare the proteomic profiles of CRC patients (*n* = 9) and found that in CRLM, FILIP1L expression was significantly lower than that in primary CRC [117]. This result suggests that CRLM cells may also have stronger EMT activity.

### 4.2. Transcription Factors

Transcription factors are a large class of proteins that specifically bind to target genes and are important parts of transcriptomic regulation [118]. Transcription factors are inextricably linked to tumors and can alter their activity in tumors and promote tumor proliferation and invasion through chromosomal mutations, gene amplifications or deletions, and point mutations [119]. For example, promyelocytic leukemia protein (PML)-retinoic acid receptor α (RARα) is a driver of leukemia, and the transcription factor yin yang 1 (YY1), whose expression is upregulated in most tumors, can regulate the pentose phosphate pathway by modulating the activity of G6PD (6-phosphogluconate dehydrogenase) and inducing proliferation and metastasis of tumor cells [120,121]. In CRC, death domain-associated protein (DAXX) is a tumor suppressor that acts as a transcriptional repressor in the nucleus and affects the progression of CRC. Vertebrate zinc finger E-box binding homeobox (ZEB) proteins are a family of transcription factors. Liu et al. found that the expression of DAXX was downregulated in CRLM compared with primary CRC [122]. As a transcriptional repressor, DAXX inhibits the expression of ZEB-mediated E-cadherin. MYC and hypoxia-inducible factor-1 (HIF1α) were increased in liver metastasis compared to their primary tumors [123]. Due to the long-term hypoxia of tumor cells, HIF1α is also in a state of high expression. The high expression of HIF1α activates downstream effector genes, and the MYC gene is essential for HIF1α to promote cell proliferation [124].

### 4.3. Other Proteins

In addition to FILIP1L, Ku et al. also identified the remaining 46 differentially expressed proteins using an ANOVA (Tukey test) and identified plasminogen (PLG), which was the most upregulated protein [117]. Plasmin is needed by CRC cells to hydrolyze the extracellular matrix, and PLG is involved in the plasminogen activation system (PAS). This indicates that the upregulation of PLG expression allows CRC cells to undergo distant metastasis through the extracellular matrix. Kim et al. performed mass spectrometry on five CRLM patients and validated the results by Western blotting (WB) [125]. Of 164 proteins, the reduced expression of 51 proteins and increased expression of 7 proteins was observed. The reduced proteins were mainly in the mitochondrial matrix, the mitochondrial intermembrane space, the proteasome complex, and the actin cytoskeleton and play a role in protein and ATP synthesis and actin dynamics. Thus, actin dynamics, protein degradation, and ATP synthesis are reduced in CRLM compared to primary CRC. In contrast, the seven proteins with increased expression were mainly serpin family A member 1 (SERPINA1), apolipoprotein A1 (APOA1A), carbonic anhydrase 1 (CA1), and succinate dehydrogenase complex flavoprotein subunit A (SDHA). Serpin A1 is a protease inhibitor that is regulated by the Snail protein and can promote tumor cell invasion and metastasis. Many studies have shown elevated levels in serum in patients with a variety of tumors, including lung and gastric cancers [126,127]. SDHA is considered a tumor suppressor and its loss of function is associated with the development of kidney cancer and breast cancer [128,129]. However, SDHA upregulation in CRLM is intriguing, and perhaps it plays an opposite role in CRC.

We summarize the genetic, transcriptomic, and protein heterogeneity of CRLM in Table 1. From Table 1, we show that the genetic heterogeneity in the above three aspects of CRLM is not significant, which suggests that these genes may play similar roles in primary CRC and CRLM, whereas the heterogeneity in CRC may be more focused on other aspects. Compared with genetic heterogeneity, transcriptome heterogeneity was more pronounced (Table 2). Furthermore, the functions of these differentially expressed RNAs varied, but most of them were involved in the EMT process of CRC cells and in their progression and metastasis. Tumor cells undergo EMT to experience distal metastasis and a large number of proteins are changed during EMT to support the transformation and metastasis of tumor cells; therefore, these cells also show significant heterogeneity after metastasis. Further study of heterogeneity at the protein level reveals a series of changes in CRLM and many contribute to CRLM by affecting the EMT process (Table 3). The expression of migration-related proteins such as VTN, ARP3, FN1, and TIMP1, which are responsible for promoting tumor cell migration and infiltration, is increased in CRLM. Meanwhile, the expression of adhesion-related proteins, such as claudins and ITA5, is decreased, which leads to the loss of tumor cells to adhere to neighboring cells and thereby increase their metastatic ability. In addition, some proteins, including transcription factors, play a role in affecting EMT-related pathways, such as the Wnt/β-catenin, TGF-β, and Notch signaling pathways that play an important role in the development and metastasis of CRLM. In conclusion, the protein heterogeneity of CRLM is mainly and closely related to the EMT process in which it participates. By understanding these changes, we can better understand the mechanisms of CRLM development and metastasis and provide new clues for developing therapeutic strategies targeting the EMT process. However, the conclusions we present are speculative and need further validation by additional basic research and clinical trials.

## 5. Metabolic Heterogeneity

Metabolic reprogramming is one of the hallmarks of cancer [130]. Compared with normal tissues, tumor cells often require more energy to maintain their growth. Due to the different microenvironments of metastatic organs, tumor cells still need to undergo metabolic reprogramming to obtain energy for growth in different metastatic organs.

Tumor cells are often in a state of aerobic glycolysis, the so-called “Warburg effect”, and even with sufficient oxygen, cells prioritize glycolysis to quickly generate energy rather than through the tricarboxylic acid cycle (TCA cycle). After CRLM, some specific growth factors and enzymes in the liver make this effect more obvious in metastatic lesions. The expression of glucose transporter 3 (GLUT3) and pyruvate kinase muscle isozyme 2 (PKM2) is also significantly higher in CRLM [131]. Increased glucose uptake mediated by GLUT3 can promote the occurrence of various tumors including liver cancer, breast cancer, and lung cancer [132,133,134]. The overexpression of GLUT3 activates Yes-associated protein (YAP), which in turn promotes the expression of GLUT3 and glycolytic genes; conversely, the expression of GLUT3 and glycolytic genes is decreased after YAP is knocked down. Meanwhile, YAP also interacts with PKM2 through the WW domain and collectively enhances the expression of GLUT3. GLUT3 and YAP/PKM2 constitute a positive feedback pathway that enhances glycolysis in CRLM [131]. Some studies have also reported the mechanism of GLUT3 upregulation in CRLM. High mobility group proteins (HMGs) are a class of structural transcription factors that do not have transcriptional activity, but they can regulate the transcription of target genes by binding with their structures. Yang et al. found that HMGA1 can promote the expression of GLUT3 in CRLM and thereby enhance the GLUT3-YAP signaling pathway [135].

Next, the expression of phosphorylated PKM2 is higher in CRLM than in primary CRC, it can act as a transcriptional cofactor for hypoxia-inducing Factor 1 (HIF-1), and it promotes the expression of glycolytic genes including LDHA, PDK1, and SLC2A1 (GLUT1) [136]. In addition to the two proteins PKM2 and GLUT3, Deng et al. also identified another differentially expressed protein, Dickkopf-associated protein 2 (DKK2), which promotes aerobic glycolysis in CRC cells [137]. By comparing the proteomes of primary CRC and CRLM from seven patients, Fahrner et al. found that most of the proteins upregulated in CRLM were involved in glucose metabolism, including pyruvate carboxylase, fructose-bisphosphate aldolase B, and fructose-1,6-bisphosphatase 1 [138]. Finally, according to Bu et al., the expression of aldolase B (ALDOB), an enzyme involved in fructose metabolism, is increased in CRLM, and overexpressed ALDOB enhances fructose metabolism and thereby generates more propanose phosphate [139]. The production of large amounts of propyl phosphate also promotes glycolysis in CRC cells.

In addition, enhanced cholesterol synthesis and upregulated expression of some fatty acids, acylcarnitines, and polyamines have also been found in CRLM [140,141]. As previously described, SREBP2 is a key transcription factor for lipid synthesis. Zhang et al. found that the expression of SREBP2 and its downstream target genes, LDLR and SRB1, were significantly upregulated in CRLM [140]. These authors subsequently knocked down SREBP2 and found that total cholesterol levels in tumor cells were significantly reduced and tumor cell growth was restricted. After screening several liver-rich growth factors, they finally found that hepatocyte growth factor (HGF) in the liver promotes the PI3K/AKT/mTOR pathway, which stimulates SREBP2 and stimulates cholesterol synthesis in CRLM [140]. Finally, Williams et al. found that several phosphatidylcholines, carnitine, bile acids, nucleotides, oxidative compounds (glutathione), and polyamines (putrescine) were expressed significantly higher in CRLM than in primary CRC [141]. Glutathione (GSH) protects cells against oxidative stress and polyamines are important growth factors needed for cell growth.

Overall, we show in Table 4 the changes in metabolic reprogramming of CRLM that allow CRC cells to adapt more quickly to the metabolic state of the liver and thereby promote their growth in the liver. From the table, we can determine that glycolysis-related heterogeneity is the most obvious, which is probably because glycolysis can generate a large amount of energy, and it provides energy in the process of CRC cell metastasis and colonization in the liver. Furthermore, the other metabolites also upregulate and promote the growth of CRC cells. From our conclusion, we identify that CRLM possesses a more active metabolic state to maintain cell growth compared to primary CRC.

## 6. Immune Heterogeneity

The immune microenvironment of tumors is a complex system, and immune cells in the microenvironment have been shown to influence tumor progression and response to immunotherapy [142,143]. Current research on the heterogeneity of the immune microenvironment is still mainly focused on immune cells. During tumor metastasis, immune cells are dynamically heterogeneous, which means that cell types, numbers, and sizes change [144].

### 6.1. Macrophages

Tumor-associated macrophages (TAMs) are closely associated with tumor progression and angiogenesis [145]. SPP1^+^ TAMs are immunosuppressive cells that are reported to be highly expressed in CRC compared to normal tissues, which can promote CRC progression and metastasis, and are also associated with the prognosis and response to immunotherapy in CRC patients [146,147]. Liu et al. found that SPP1^+^ TAMs are malignancy-associated and are linked to CRLM [148]. The angiogenesis and phagocytosis properties of three types of TAMs, MKI67^+^ TAMs, SPP1^+^ TAMs, and C1QC^+^ TAMs were also compared in the context of CRC. Results revealed that SPP1^+^ TAMs possessed the strongest angiogenic function, which confirmed their immunosuppressive and protumorigenic functions. Wu et al. used single-cell RNA sequencing and spatial transcriptomics to determine 97 CRC-paired samples and derived a single-cell spatial map of CRLM [6]. Results demonstrated that MRC1^+^ CCL18^+^ TAMs, SPP1^+^ TAMs, and neutrophils were significantly increased in CRLM compared to matched primary CRC. Neutrophils are reported to be potential tumor-promoting cells [149]. Wu et al. focused their research on MRC1^+^ CCL18^+^ TAMs. They suggested that MRC1^+^ CCL18^+^ TAMs might originate from Kupffer cells in the liver and found that M2 polarization-related genes (APOE, MARCO) were significantly upregulated in MRC1^+^ CCL18^+^ TAMs of CRLM, while MRC1^+^ CCL18^+^ TAMs of primary CRC showed higher expression of inflammatory cytokines (TNF, IL1B, CCL3, and CCL4). Additionally, they found that the MRC1^+^ CCL18^+^ TAMs of CRLM possessed strong metabolic activity, mainly in terms of phenylalanine metabolism, whereas the MRC1^+^ CCL18^+^ TAMs of CRC were dominated by oxidative phosphorylation. Moreover, both SPP1^+^ TAMs and MRC1^+^ CCL18^+^ TAMs showed enhanced antigen processing and presentation and complex activity. Tu et al. also found more TAM enrichment in CRLM and dominance of M2 TAMs [150]. This phenomenon was associated with the elevated expression of TCF4 in CRLM. TCF4, a transcription factor involved in the WNT/TCF signaling pathway, recruits TAMs and promotes TAM M2 polarization mainly by promoting the expression of two monocyte chemokines, CCL2 and CCR2.

In addition, earlier studies have shown that macrophages are morphologically heterogeneous. For example, M1-like macrophages are often round or flat, whereas M2-like macrophages are elongated [151,152], and macrophages acquire different geometries in different tissues [153,154]. Therefore, it is conceivable that during CRC metastasis to the liver, macrophages change not only in type and gene expression but also in morphology. Donadon et al. investigated this phenomenon and found a significant increase in the area and circumference of macrophages in CRLM, which they termed large (L-TAMs) macrophages [155]. These L-TAMs have a strong lipid-metabolizing capacity, while inflammation-related pathways (leukocyte extravasation, acute phase response, and NF-κB signaling) are downregulated. Finally, both complement-related pathways and their genes were highly expressed in these L-TAMs, which is a result that is consistent with the findings of Wu [6].

### 6.2. T Cells

T cells play an important role in tumor progression and metastasis. Cytotoxic T cells can secrete granzyme and perforin to kill tumor cells, while regulatory T cells (Tregs) can suppress the immune response and promote tumor cell development [156]. During tumor progression, large numbers of CD4^+^ T cells and CD8^+^ T cells are usually depleted, and studies have demonstrated that during the course of CRLM, the numbers of these two cells are significantly reduced and more CD4^+^FOXP3^+^ Tregs are found in liver metastasis [157,158]. Another study found that T helper cell (Th)17 as well as two other Tregs (Treg-IL10 and Treg-CTLA4) are enriched in primary CRC, and furthermore, the proportions of Treg-IL10 and Treg-CTLA4 in primary foci of CRC patients with the presence of liver metastases were significantly higher compared to non-metastatic CRC tumors [148]. Treg-IL10 show high expression of IL10, IL23, and IL1R1. The role of IL10 in tumors is controversial, as on the one hand, it can inhibit the function of antigen-presenting cells and block T-cell killing function against tumors, while on the other hand, it can inhibit angiogenic factors and activate CD8^+^ T cells [159,160]. The chronic inflammation-associated cytokine IL23 can promote tumor progression [161]. Treg-CTLA4 is highly expressed in Treg activation-related factors, including LAYN, CCR8, and TIGIT [148].

### 6.3. Dendritic Cells

Dendritic cells (DCs) can be divided into plasmacytoid DCs (pDCs) and conventional DCs (cDCs) in both humans and mice [162]. cDCs can be further divided into two phenotypically and functionally distinct subsets. cDC1s express Toll-like receptors (TLRs) and secrete proinflammatory cytokines, including IL-12p70 and IFN-α, to induce Th1 responses. cDC2 mainly acts as an antigen-presenting cell and activates effector T cells, including Th2 and Th17 cells [162,163]. Presently, there are few studies on the heterogeneity of dendritic cells between primary CRC and CRLM. Liu et al. identified 10 DC subsets in CRLM patients and found great heterogeneity in two types of cDC2s (cDC2-C1QC and cDC2-TIMP1) [148]. Because cDC2-C1QC was highly expressed in C1QA, CD68, CD163, and CD14, similar to the recently identified DC3 population [164,165], it was identified as DC3s. Overall, cDC2-C1QC cells showed a higher proinflammatory profile, whereas cDC2-TIMP1 cells exhibited a high expression of maturation markers (such as CCR7) and angiogenesis-related genes (EREG, CREM, and VEGFA). In addition, cDC2-TIMP1 cells expressed more anti-inflammatory genes than DC3 cells. These results revealed that cDC2-C1QC was enriched in CRC in contrast to cDC2-TIMP1, which was more abundant in CRLM.

Immune cells play a major role in the immune response to tumors, and different types of immune cells either kill tumor cells or promote their immune escape. In Table 5, we summarize the types of cells with the most pronounced heterogeneity in CRLM and immunosuppressive cells occupy the majority. Compared to CRC, the accumulation of these immunosuppressive cells in the liver increases the probability of immune escape from the tumor and reduces the response of CRLM to immunotherapy.

## 7. Discussion

Due to the complex chain reaction in vivo, tumor progression and metastasis involve changes in many biological processes and their corresponding factors. There are currently two models that can be used to explain this variation (i.e., tumor heterogeneity): the clonal evolution model and the cancer stem cell model. The cancer stem cell model suggests that cancer stem cells (CSCs) can differentiate into different types of cells, thus becoming a major source of intratumor heterogeneity [166]. CRC has a high incidence and often metastasizes to the liver. Based on the cancer stem cell model, we hypothesized that during CRLM, some of the CSCs from the primary tumor metastasized to the liver and then differentiated into different cellular subpopulations, resulting in significant heterogeneity from the primary focus. As mentioned above, the generation of heterogeneity results in CRC primary foci and CRLM showing large differences in response to treatment, so understanding the heterogeneity between primary CRC and corresponding CRLM becomes particularly important. First, it can provide a theoretical basis for explaining such differences in treatment. Second, these differentially expressed genes or proteins, etc., can also serve as new molecular markers for the study of treatment, opening up a new direction for clinical treatment. In this review, we summarize the heterogeneity between primary CRC and CRLM at the genetic, transcriptomic, protein, metabolic, and immune levels.

As shown in Table 1, Table 2 and Table 3, heterogeneity is evident at all four levels except for genetic heterogeneity. Although the differentially expressed noncoding RNAs, transcription factors, and proteins are diverse and each performs different functions, the vast majority of them promote CRC progression and metastasis by promoting EMT and angiogenesis (Figure 1). Enhanced glycolysis, fatty acid synthesis, and other processes also provide liver metastatic cells with more energy to sustain growth. In addition, a decrease in the number of cytotoxic cells, such as CD4^+^ T cells and CD8^+^ T cells, can also be found in the liver, which is replaced by more immunosuppressive cells, such as Tregs and TAMs (Figure 2).

There is strong evidence that intratumor heterogeneity is a major challenge in the clinical treatment of oncology patients. Thus, as shown in Table 1, Table 2 and Table 3, the heterogeneity reflected in these five levels may serve as new molecular biological markers and new targets for the treatment of CRC patients and provide new therapeutic directions for the clinical treatment of CRLM patients.

First, and as previously mentioned, CSCs may be a major source of tumor heterogeneity. CSCs are cells with self-renewal and pluripotent differentiation capabilities and are considered the main drivers of tumor growth, metastasis, and recurrence [167]. Due to their unique biological properties, CSCs are highly resistant to conventional chemotherapy and radiotherapy methods [168]. Thus, CRLM patients may benefit from therapies targeting CSCs [169]. Li et al. found that METTL3 could maintain the phenotype of CSCs, and despite the lack of clinical trials, it has been demonstrated through in vitro experiments that the knockdown of METTL3 reduced the frequency of colorectal CSCs, and knocking down METTL3 in the PDX model also reduced the size of the tumors [95]. Therefore, in subsequent therapeutic studies targeting CSCs, targeting METTL3 may be a good choice.

Second, targeting the EMT process in CRC is also a hot topic in current clinical research, and targeting EMT-related genes, RNAs and proteins is one way to inhibit EMT. Zhang et al. detailed a summary of current drugs that target EMT, such as fresolimumab, a monoclonal antibody targeting TGF-β, and regorafenib, which targets factors such as BRAF and VEGF [170]. From this summary, it is clear that current research in this area is focused on factors such as TGF-β, AKT, EGFR, and HGM2. We summarized the specific biomarkers involved in the EMT process of CRLM, which provides new directions for the development of additional EMT inhibitors.

Targeted metabolic approaches have also been shown to be highly effective in cancer treatment, including targeting aerobic glycolysis to inhibit glucose uptake by tumor cells and targeting fatty acid synthesis and amino acid metabolism [171]. In fact, a number of drugs targeting metabolism have been approved for clinical use, especially those targeting nucleotide metabolism, such as the FOLFIRI regimen (a regimen consisting of three drugs: fluorouracil, leucovorin, and irinotecan), which has already proven to be effective in the treatment of CRC and can also be used in conjunction with chemotherapy and antiangiogenic agents to obtain better therapeutic results [171,172]. Therefore, targeted metabolism is a very promising therapeutic area, and proposing new targeted metabolisms may have a very significant impact on clinical therapy. Bu et al. provided a new direction by targeting fructose metabolism [139]. Similarly, increased GSH in CRLM is associated with tumor progression and drug resistance. Increased GSH can lead to drug resistance in CRC cells by binding to drugs, interacting with reactive oxygen species, preventing protein or DNA damage, or participating in DNA repair processes [173,174]. Therefore, GSH is a potential therapeutic target for CRC patients. Studies have also demonstrated that GSH depletion therapy combined with reactive oxygen species-based therapy (photodynamic therapy (PDT), sonodynamic therapy (SDT), and chemodynamic therapy (CDT)) may improve the therapeutic effect for CRLM patients [175].

Finally, the liver has a unique immunosuppressive environment [176]. Table 3 shows that CRLM contains more immunosuppressive macrophages, and the infiltration of CD8^+^ T cells and CD4^+^ T cells is also significantly reduced, which indicates that liver metastasis aggravates the immunosuppressive microenvironment of the liver and reduces the antitumor immune response. Thus, the application effect of immune checkpoint inhibitors (such as PD-1/PD-L1 inhibitors) in patients with CRLM is weakened. In a recent study, a preclinical model was developed, and in the presence of liver tumors, an increase in Tregs in the liver caused changes in tumor antigen-specific myeloid-derived suppressor cells (MDSCs) in distant subcutaneous tumors and ultimately suppressed the activation of CD8+ T cells which resulted in suppression of the tumor immune response. In other words, liver tumors may lead to a reduction in the effectiveness of systemic anti-immunotherapy. To address this situation, preclinical trials of anti-CTLA-4 mAb and the EZH2 inhibitor CPI-1205 should be conducted in liver metastasis cases, as both drugs inhibit Tregs and improve the immune microenvironment in the liver or in distant primary foci [177]. In addition, Wu et al. found that neoadjuvant chemotherapy (NAC) can inhibit the activity of MRC1^+^ CCL18^+^ TAMs and SPP1^+^ TAMs and thereby enhance the response of CRLM to immunotherapy. These authors also proposed new clinical combination regimens such as combining NAC, metabolism checkpoint inhibitors, and immunotherapy [6]. Targeting these immunosuppressive cells in combination with immunotherapy may become the primary treatment modality for patients with CRLM in the future.

## 8. Conclusions

In conclusion, we summarized the intratumor heterogeneity between primary CRC and CRLM in this review. Overall, the clinical effects of these heterogeneities need further investigation and may hold promise as new targets for the treatment of CRLM patients.

## Figures and Tables

**Figure 1 cancers-16-00059-f001:**
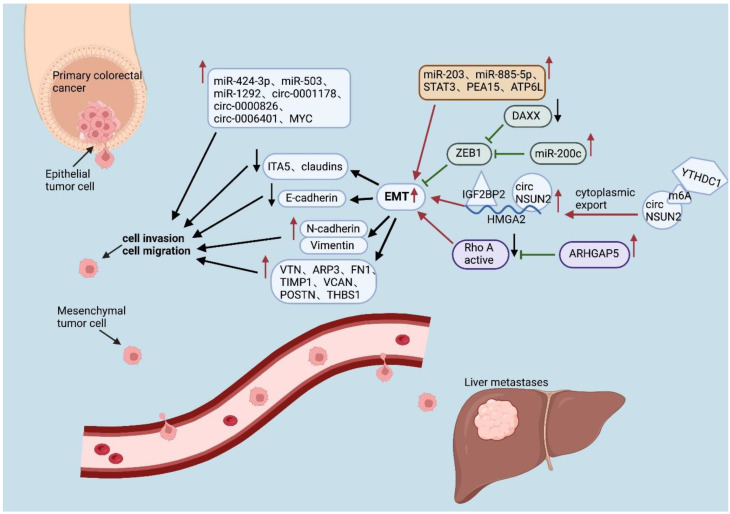
Different noncoding RNAs, transcription factors, and proteins affect EMT processes in colorectal cancer cells through different mechanisms. (↑: up-regulate; ↓: down-regulate; ←: promote; ⊢: inhibit).

**Figure 2 cancers-16-00059-f002:**
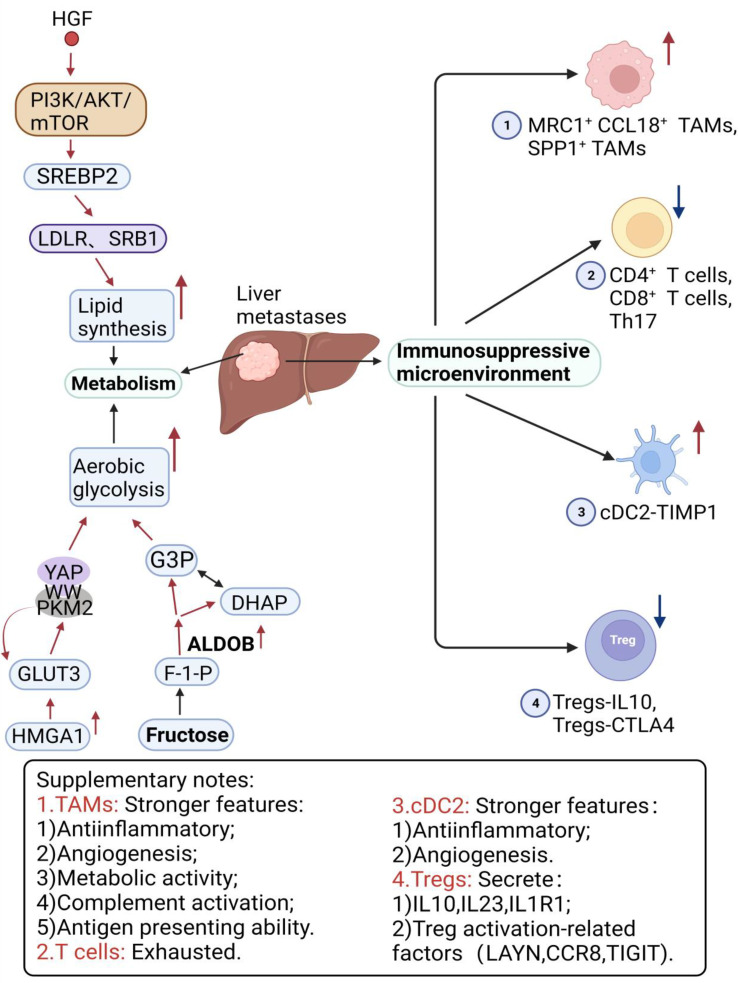
The enhanced metabolic and immunosuppressive microenvironment in the liver provides better survival conditions for CRLM. (↑: up-regulate; ↓: down-regulate).

**Table 1 cancers-16-00059-t001:** Summary of genetic heterogeneity in CRLM (↑: upregulated, -: no change).

Items	Factors	Change	References
APC, RAS, BRAF, PIK3CA, TP53, SMAD4, MSI status		-	[21,22,23,24,25,26,27,30,43,44,45,46]
DNA copy numbers		-/↑	[34,35,36,37,40,41]

**Table 2 cancers-16-00059-t002:** Summary of transcriptomic heterogeneity in CRLM (↑: upregulated, ↓: downregulated).

Items	Factors	Change	References
MiRNAs	MiR-122, MiR-122*, MiR-885-5p, MiR-203, MiR-200c, MiR-424-3p, MiR-503, MiR-1292	↑	[51,52,53,60,61,65]
	MiR-143, MiR-10b, MiR-28-5p	↓	[51,52,53]
CircRNAs	Circ0001178, Circ0000826, CircNSUN2, Circ0006401	↑	[71,72,73,74,75,76,77,78,79,80,81,82,83]
LncRNAs	LncRNA GAL, LncRNA MIR17HG	↑	[86,87]
DNA methylation	RASSF1a, CircNSUN2, METTL3	↑	[81,94,95]
	p16INK4a	↓	[94]
Protein acetylation	H2BLK121Ac, ADH1B K331Ac, IDH1	↑	[96,97]
	HIST2H3AK19Ac, TPM2 K152Ac	↓	[96]

**Table 3 cancers-16-00059-t003:** Summary of protein heterogeneity in CRLM (↑: upregulated, ↓: downregulated).

Items	Factors	Change	References
EMT-related proteins	Migration-associated proteins: VTN, ARP3, FN1, TIMP1, VCAN, POSTN, THBS1	↑	[99,100]
	IGFBP7, Adhesion protein: claudins, ITA5	↓	[99,106,107,108,109,110]
	ARHGAP5, PEA15, ATP6L, FILIP1L	↑	[111,114,117]
Transcription factors	MYC, HIF1α	↑	[123]
	DAXX	↓	[122]
Other proteins	PLG, Serpin A1, APOA1A, CA1, SDHA	↑	[114,125]
	the mitochondrial matrix, the mitochondrial intermembrane space, the proteasome complex, and the actin cytoskeleton	↓	[125]

**Table 4 cancers-16-00059-t004:** Summary of metabolic heterogeneity in CRLM (↑: upregulated).

Items	Factors	Change	References
Aerobic glycolysis	GLUT3, HMGA1, PKM2, DKK2, Pyruvate carboxylase, Fructose-bisphosphate aldolase B, Fructose-1,6-bisphosphatase 1	↑	[131,135,136,137,138]
Fructose metabolism	ALDOB	↑	[139]
Cholesterol metabolism	SREBP2, LDLR, SRB1	↑	[140]
Fatty acids, acylcarnitines, oxidative compounds, polyamines	GSH, putrescine	↑	[141]

**Table 5 cancers-16-00059-t005:** Summary of immune heterogeneity in CRLM (↑: upregulated, ↓: downregulated).

Items	Factors	Change	References
TAMs	MRC1^+^ CCL18^+^ TAMs, SPP1^+^ TAMs	↑	[6,148,150]
T cells	CD4^+^ FOXP3^+^ Tregs	↑	[157,158]
	Th17, CD4^+^ T cells, CD8+ T cells, Treg-IL10, Treg-CTLA4	↓	[148,157,158]
DCs	cDC2-TIMP1	↑	[148]
	cDC2-C1QC(DC3s)	↓	[148]
Neutrophils		↑	[6]

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
