# Peer review of "Multilevel Heterogeneity of Colorectal Cancer Liver Metastasis"

_cancers, 2023, doi:10.3390/cancers16010059_

Round 1

Reviewer 1 Report

Comments and Suggestions for Authors

The literature review by Chen et al. describes different levels of heterogeneity for Colorectal Cancer Liver Metastasis (CRLM). Specifically, the authors discuss five types of heterogeneity: 1) genetic, 2) transcriptomic, 3) protein, 4) metabolic, and 5) immune. Overall, the manuscript is fairly well-written and presented in a logical order, and the number and content of the figures and tables are appropriate. There are some recommendations on what to add and reorganize to improve the overall comprehensiveness. Also, there are a number of minor points to address for clarity.

Major comments: 

1. English editing is required to improve the flow, especially for the Abstract and Introduction.

2. Epigenetic heterogeneity needs to be expanded. The authors should add a section on epigenetic regulation of gene expression, including DNA methylation, promoter regulation, CIMP, and histone modifications. This could be included in section 3. “Transcriptomic heterogeneity”. Also adding a short introductory paragraph to this section is recommended.

3. The inclusion of transcription factors in section 3 “Transcriptomic heterogeneity” is confusing, because of course, transcription factors are proteins.  Thus, there is also a lot of overlap with the following section 4 “Protein heterogeneity” eg. Snail, Twist, Wnt/beta-catenin. It would be more logical to move the discussion on transcription factors and incorporate this into the Protein Heterogeneity section.

4. Discussion of cancer stem cells/cancer-initiating cells should be included where appropriate, and therapeutic implications.

5. For the Tables, inclusion of the first column “levels” is redundant with the title and should be omitted (all Tables).

6. Figure 2 is for the difficult to read due to its small size, especially the small font. This should be re-done – enlarged and reduce amount of text if possible.

Minor comments: 

1. The author callouts are not formatted correctly (should be single name)

2. CRC is a disease not a “tumor” (line 30).

3. Use the proper gene name for PIK3CA throughout (section 2.1 Key driver genes)

4. Mutations in the SMAD4/TGFBR2 pathway should be included in the genetic heterogeneity

5. When comparing primary CRC tumors to liver metastases, it should be written as “primary CRC” versus CRLM, rather than just “CRC”, which is ambiguous.

6. For both circRNAs and lncRNAs, add a sentence defining size and how each regulates gene expression at the molecular level.

Comments on the Quality of English Language

English editing is required to improve the flow, especially for the Abstract and Introduction.

Author Response

Dear Reviewer 1

  We feel great thanks for your professional review work on our article. As you are concerned, several problems need to be addressed. According to your nice suggestions, we have made extensive corrections to our previous draft, the detailed corrections are listed below. Please see the attachment.

  Thank you again for your positive comments and valuable suggestions.

  Best regards,

  Mr. Hao Chen

Reviewer 2 Report

Comments and Suggestions for Authors

Nice coprehensive review. Just minor language editing should be advisable. In addition, comma sign is wrong in figures and this should be changed.

Author Response

Dear Reviewer 2

  We feel great thanks for your professional review work on our article. As you are concerned, several problems need to be addressed. According to your nice suggestions, we have made extensive corrections to our previous draft, the detailed corrections are listed below. Please see the attachment.

  Thank you again for your positive comments and valuable suggestions.

  Best regards,

  Mr. Hao Chen

Reviewer 3 Report

Comments and Suggestions for Authors

Overall a well written paper with an organized article layout. Unfortunately some of the conclusions are inaccurate. For example, "These drugs can improve the prognosis of CRLM patients when used as adjuvant therapy or chemopreventive agents." when referring to regorafenib and fresolimumab. This is very speculative as to my knowledge neither of these have shown benefit in these contexts in a clinical trial. (reference is a review which does not extensively discuss these.)

Another example is: "A recent study also found that this immunosuppressive effect of the liver is achieved by Tregs, and CTLA-4 inhibitors can inhibit the effect of Tregs, so the combination of CTLA-4 inhibitors and PD-1 inhibitors has a significantly better antitumor effect than PD-1 inhibitors alone." This is true in animal models but has not held up in clinical trials unfortunately. 

Author Response

Dear Reviewer 3

  We feel great thanks for your professional review work on our article. As you are concerned, several problems need to be addressed. According to your nice suggestions, we have made extensive corrections to our previous draft, the detailed corrections are listed below. Please see the attachment.

  Thank you again for your positive comments and valuable suggestions.

  Best regards,

  Mr. Hao Chen

Reviewer 4 Report

Comments and Suggestions for Authors

In this review paper, the authors presented the main trends in investigations of heterogeneity of colorectal cancer and сolorectal сancer liver metastasis. The topic is very interesting. The material is described in sufficient detail, but there are several comments:

1.    Is there any data in the literature about epigenetic changes in CRLM?

2.    May be it is worth combining tables 1, 2 and 3. And make title of this table for example “Nucleic acids and protein level…”

3.    There are a lot of old links in this review it is worth adding from articles over the past years.

Author Response

Dear Reviewer 4

  We feel great thanks for your professional review work on our article. As you are concerned, several problems need to be addressed. According to your nice suggestions, we have made extensive corrections to our previous draft, the detailed corrections are listed below. Please see the attachment.

  Thank you again for your positive comments and valuable suggestions.

  Best regards,

  Mr. Hao Chen

Round 2

Reviewer 1 Report

Comments and Suggestions for Authors

Overall, the revised manuscript is improved from the original version with expanded scientific content and clarity. However, there are some minor points to address in the revised manuscript:

1. The rewriting of the Simple Summary was unnecessary and unrequested, as far as I can tell from the Reviewer comments.  In addition, the structure of this new paragraph is very poor, and includes multiple run-on sentences and a lot of repetition.  The authors should revert to the Simple Summary in the original manuscript.

2. CRC is a disease not a “tumor” (line 19).

3. Revised Table 1 is missing and could not be evaluated.  In addition, I respectfully disagree with the suggestion of the other reviewer to maybe combine the first three original tables into one.  I think it is better to have one table per level of heterogeneity described, therefore five tables are appropriate.

4. Regarding Figure 2, the original comment was not addressed, which is that will be difficult to read at publication size due to its small size, especially the small font. The author’s response was to make the entire figure larger and change the orientation of the page, to a size and format at which it would likely not be published.  Again, the content of Figure 2 should be re-done – enlarged and reduce amount of text if possible.  Perhaps it could be redrawn so that the height is greater than the width, thus allowing it better to fit on a standard page format.

Comments on the Quality of English Language

English editing is still required.  For example, in the Abstract and Introduction, there are run-on sentences, and issues with word order, verb tense, word choice, and repetition.  The Simple Summary in the revised manuscript is poorly written.

Author Response

Dear Reviewer 1:

Once again, we feel great thanks for your professional review work on our article. As you are concerned, there are still several problems that need to be addressed. Your comments are listed below in red italics, with specific questions numbered. Our responses are given in normal black font, and changes to the manuscript are indicated by a yellow background in black font.

  Thank you again for your positive comments and valuable suggestions.

  Best regards,

  Mr. Hao Chen

  • Comments(1): The rewriting of the Simple Summary was unnecessary and unrequested.

Response: Thank you for double-checking, and we apologize for this part of the modification. In the previous revision, the previous “simple summary” did not meet the word count requirement according to the editor's request, so we rewrote this section to meet the word count requirement (150-200 words). Based on your comments, we have revised this section again to make the language in this section more accurate.

  • Comments(2): CRC is a disease not a “tumor” (line 19).

Response: We feel sorry for our carelessness. Line 21 has been changed from “tumor” to “disease”.

  • Comments(3): Revised Table 1 is missing and could not be evaluated.

Response: We sincerely appreciate your valuable comments. We have reorganized Table 1 into three tables corresponding to genetic heterogeneity, transcriptomic heterogeneity, and protein heterogeneity.

  • Comments(4): Regarding Figure 2, the original comment was not addressed, which is that will be difficult to read at publication size due to its small size, especially the small font.

Response: Thanks to your suggestions, we reworked Figure 2, first we changed the pages to portrait orientation to meet publication requirements. Then we removed the text from the figure and added separate annotations below to make the figure more concise and the text clearer.

  • Comments(5): English editing.

Response: Thank you for the suggestion. Once again, we have invited a native English speaker from the United States to help embellish our article. We hope the revised manuscript will be acceptable to you.